# Remeshing flexible membranes under the control of free energy

Xinxin Wang[1,2], Gaudenz Danuser[1,2]*

**1** Lyda Hill Department of Bioinformatics, UT Southwestern Medical Center, Dallas, Texas, United States of America, **2** Department of Cell Biology, UT Southwestern Medical Center, Dallas, Texas, United States of America

* gaudenz.danuser@utsouthwestern.edu

## Abstract

Cell membranes are flexible and often undergo large-scale morphological changes during processes like mitosis, protrusion and retraction, or vesicle fusion. Mathematical modeling of cell membranes depends on a representation of the free-form surface by discrete meshes. During morphological changes, these meshes must be adjusted under the minimization of the total free energy. Current methodology for meshing is limited in one of two ways: 1) Free energy-dependent methods have no restriction on the mesh geometry. The resulting irregular meshes cause artifacts in follow-up models of morphodynamics. 2) Geometry-dependent methods maintain mesh quality but violate the physics of free energy minimization. To fill this gap, we regulate mesh geometries via a free-energy-determined remeshing process: adding and removing mesh elements upon morphological changes based on barrier crossings in a double-barrier potential between neighboring vertices in the meshes. We test the method's robustness by reproducing the morphodynamics of red blood cells and vesicle fusions; and we demonstrate the method's adaptability by simulating the formation of filopodia, lamellipodia and invaginations. Finally, we use the method to study a mechanical decoupling effect of two connected membrane tethers that has been recently observed experimentally, but has not been mechanistically explained in the context of a complete membrane surface. We propose a biophysical model that strengthens the decoupling effect and broadens the original interpretation of the experiment. The method is developed in C/Matlab and distributed via https://github.com/DanuserLab/biophysicsModels.

## Author summary

Many cellular functions require morphological features such as tubulations, protrusions, and invaginations. These features emerge from membranes reshaped by curvature-inducing proteins, osmotic pressure, and other mechanical factors. To elucidate these morphodynamics mathematical models describing the relation between morphology, mechanics, and molecular compoisition are indispensable. The key to such models is discretizing the membrane. One popular discretization relies on meshing membranes into fully connected triangles. When necessary, the mesh configuration is updated via remeshing operations.

**Data Availability Statement:** All relevant data are within the manuscript and its Supporting information files.

**Funding:** GD R35GM135428 National Institutes of Health https://www.nih.gov/ The funders had no role in study design, data collection and analysis,

decision to publish, or preparation of the
manuscript.

**Competing interests:** The authors have declared
that no competing interests exist.

These include adjusting the number of meshes, rearranging the mesh-to-mesh connectivity, and keeping the individual triangles nearly equilateral. Such flexibility and geometry is necessary accuracy wise. Current remeshing algorithms lack geometrical consistency or rely on user-defined rules that defy physical laws. Here, we propose a new algorithm that marries the needs for geometrical and physical accuracy in a double-barrier potential $V_{in}$. The two barriers confine the edges in the triangles to ensure their geometrical consistency. Large enough mechanical perturbations cause barrier crossings that trigger remeshing to restore the perturbed geometry. This physics-based algorithm is robust and more efficient than rule-based algorithms. We implement the algorithm to simulate distinct morphologies and to examine how lipid diffusion controls the mechanical coupling of distant locations on the membrane.

This is a *PLOS Computational Biology* Methods paper.

## Introduction

Studying the mechanics of cell membranes is a primary task in biophysical analyses of processes like morphogenesis [1], cell migration [2], and cytokinesis [3]. During these processes, the membrane adopts a wide range of shapes in response to cell-external and -internal forces. Recent advances in experimental approaches have enabled the measurement of variables that govern the relation between membrane mechanics and morphology. For example, membrane tension can be measured by optical-tweezer- or fluorescence-based approaches [4]; and membrane morphology can be measured by 3D light-sheet microscopy combined with computer vision [5].

Mathematical modeling is a critical complement to these experiments. Such models can overcome limitations in the spatiotemporal resolution of the experiment, infer unmeasured variables, and support mechanistic interpretation. Most of these models are grounded in the formalism of the Helfrich free energy [6], which describes the resistance of the membrane to bending. In addition, the total free energy of the membrane should reflect the influence of membrane-internal tension, osmotic pressure and external forces. The minimization of the sum of these energy terms defines the dynamics and final morphology of the membrane [7].

Developing membrane models to study cell morphodynamics requires a sampling of the total free energy in a discretized representation of the free form surface. Using algorithms in computer graphics, such as Delaunay triangulation [8], enables the construction of meshes of piecewise linear elements to represent the initial cell morphology in a discrete format. However, when minimizing the total free energy, the vertices in the initial mesh must be moved. Suppose the connectivity of the meshes is fixed; such motion can cause geometric distortions that limit the accuracy of the subsequent computation of curvature and free energy (Fig 1) and hence the ensuing balance of mechanical forces that dictates the morphodynamics.

### Limitations of current approaches

In the context of membrane modeling, two classes of remeshing approaches have been described. Both suffer significant limitations for the simulation of morphodynamic processes:

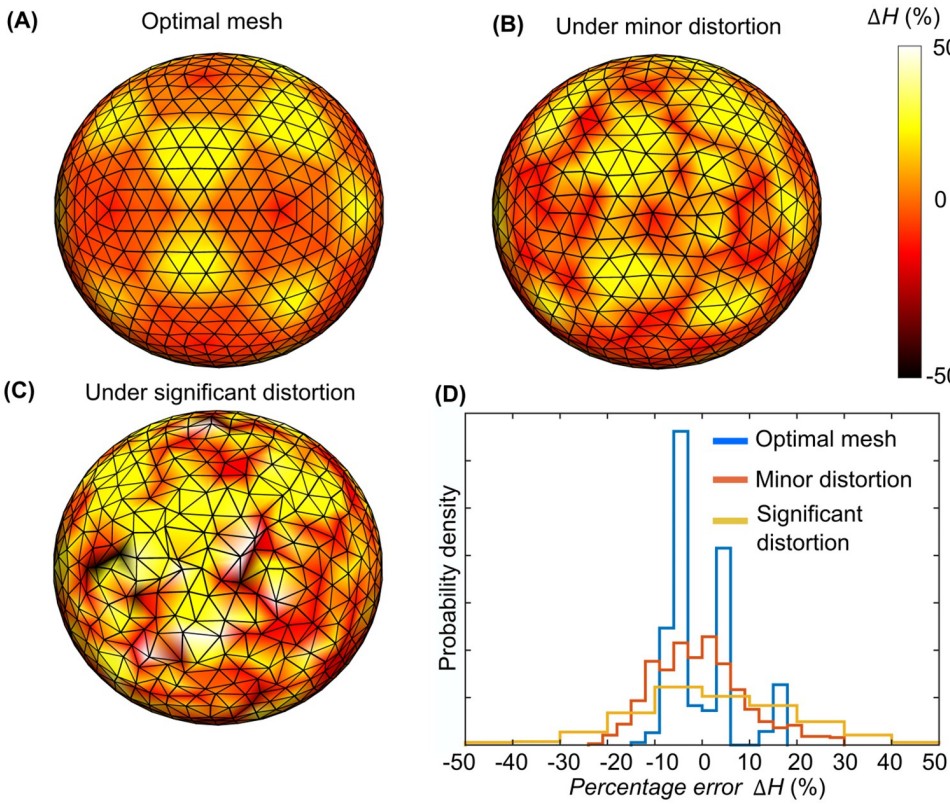

**Fig 1. Error in Helfrich energy computation on meshes with variable quality.** (A) Percentage difference $\Delta H$ between mesh calculated Helfrich energy and theoretical Helfrich energy on a sphere with optimal mesh quality (mesh generated by using [57]). (B) $\Delta H$ under minor distortion to the mesh. (C) $\Delta H$ under significant distortion. (D) Histogram of $\Delta H$ for (A-C). Percentage error $\Delta H = [\mathbf{H}(i) - H(i)]/H(i) \times 100\%$ comparing the computed, discrete value of the Helfrich energy $\mathbf{H}(i)$ at each vertex $i$ (see Methods) to the exact value $H(i) = 8\pi\kappa/N_v$, where $8\pi\kappa$ is the total Helfrich energy of an ideal sphere and $N_v$ is the number of vertices.

I) Free energy-determined approaches lack appropriate control over the mesh geometry, including flipping-based (Fig 2A) [9] and splitting- and merging-based (Fig 2B) [10] remeshing. These manipulations act on randomly chosen edges following the Metropolis Hastings scheme [9]. Any manipulation that lowers the designated energy function, such as the Helfrich free energy, is considered valid. Acceptance of other manipulations is penalized as a function of the associated energy increase. Usually, the acceptance rate of such random attempts is low, rendering these approaches computationally inefficient. More importantly, these manipulations do not always preserve mesh quality because no geometric restriction is considered. The modeled membrane is globally flexible, but maybe locally distorted. For example, as shown in Fig 2A, flipping without geometric restriction may not improve the mesh quality and often even worsen it. As a result, the computation of the free energy and the ensuing morphodynamics can be significantly distorted.

*Works related to this approach.* Boal and Seifert used flexible meshes to represent the membrane of red blood cells [11]. Random attempts of flipping were performed to identify operations that lowered the free energy of the membrane. Thermal annealing was allowed based on the Metropolis scheme: new meshes based on the flipping attempts were 100% accepted if the energy was lowered; and conditionally accepted at less and less probability if the free energy increased. The mesh quality was not precisely controlled after the flipping attempts. Sadeghi

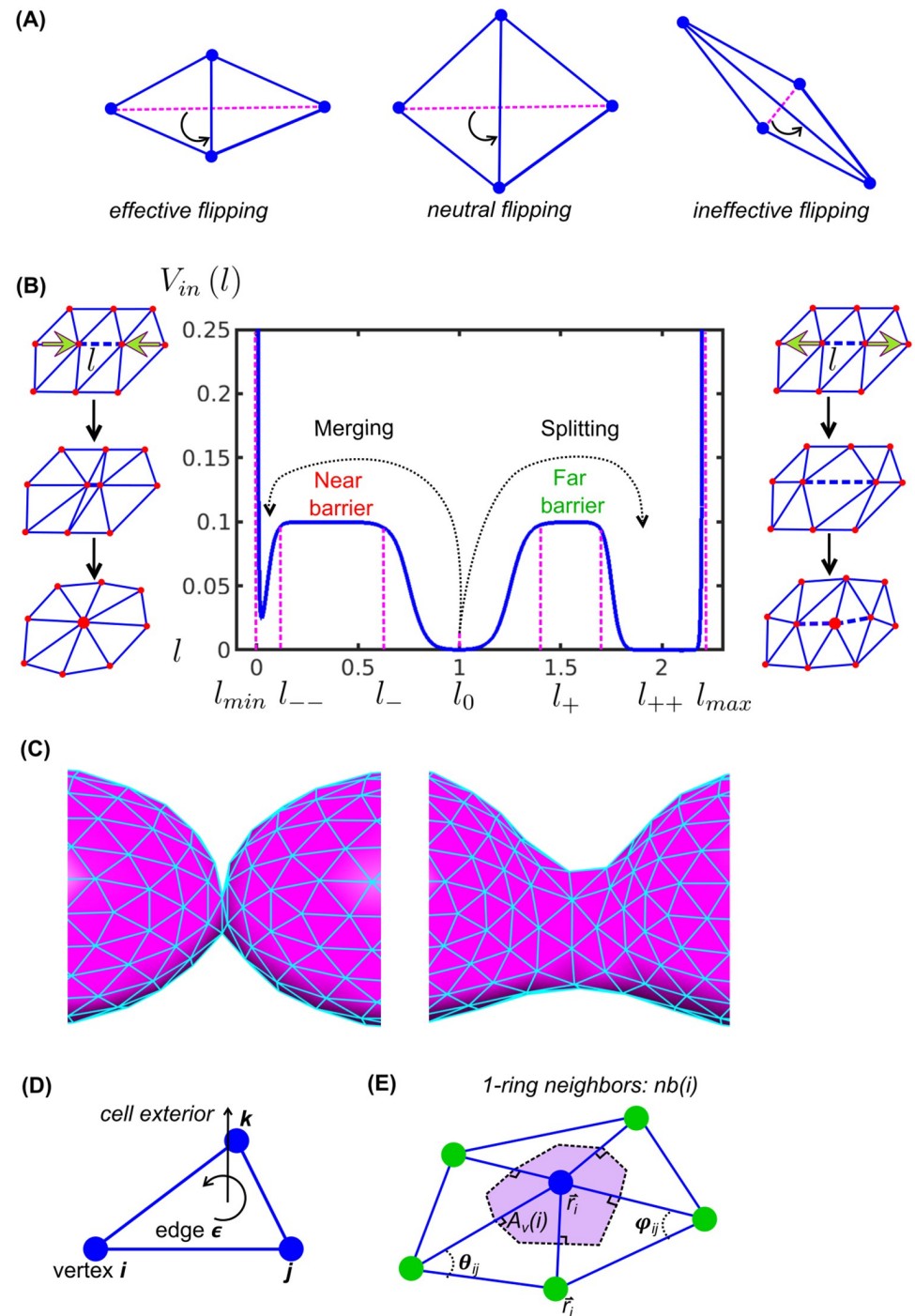

**Fig 2. Remeshing controlled by barrier crossings in a double-barrier potential.** (A) Remeshing via flipping procedures. (B) Remeshing via splitting and merging procedures controlled by barrier crossings in a potential $V_{in}$ at indicated lengths. (C) Initial triangular meshes and high-quality triangular mesh after relaxation. (D) Triangular mesh element: right-hand rule. (E) Definition of 1-ring neighbors $nb(i)$ and Voronoi area $A_v$ in purple [30]. $r_i$ and $r_j$ are the vectors defined by vertex $i$ and $j$, and $\theta_{ij}$ and $\phi_{ij}$ are the two angles for computing the discrete Helfrich energy at $i$ Eq (5).

*et al.* applied the Metropolis flipping method to simulate the dynamic process of a membrane wrapping around a spherical particle [12]. Giani *et al.* applied the Metropolis flipping method to simulate membrane deformation during endocytosis [13]. For additional variants of flipping, we refer to the review [9]. Other than flipping, Hoppe *et al.* implemented splitting and merging as an energy lowering process while maintaining the topological features of the original meshes [10]. Ma and Klug [14] introduced an artificial viscosity term to achieve r-adaptive remeshing [15] that kept the connectivity of the edges and the equilateral shape of the triangles but allowed area variation among individual triangles. Other related works include [16–18]. Despite the effectiveness in describing global membrane deformations, these approaches all suffer from local mesh distortions that deteriorate curvature-dependent computation of free energy.

II) Geometry-based manipulations restore triangles with an unfavorable edge or angle to nearly equilateral [19]. For instance, long edges are split and vertices delimiting too short edges are merged iteratively until all edges are within the desired range to achieve sufficient mesh quality [20]. Although effective at mesh-quality control, the underlying rules are arbitrary and lack the connection to free energy minimization. Thus, they are incompatible with physically realistic simulations of morphodynamics.

*Works related to this approach.* Botsch and Kobbelt used the maximal and minimal length of edges to trigger splitting and merging [20]. Flipping was performed to reduce the variation in the number of neighboring vertices. Dunyach *et al.* applied the rules in [20] to adaptively remesh the triangles [21], so that they stayed nearly equilateral while allowing them to vary in size. As a result, regions with high curvature variation were meshed more finely compared to regions with low curvature variation. Ivrissimtzis *et al.* used a neural network based algorithm to achieve high-quality meshing [22]. Guo and Hai used tightly packed spheres and adjusted their radius, so that connecting the centers of every three nearby spheres yielded high-quality triangles [23]. We refer to [19, 24, 25] for more comprehensive reviews of this strategy. Despite the effectiveness in maintaining regular meshes, these approaches rely on empirical rules that are unrelated to the physics of membrane deformation.

## Contribution

The main contribution of this work is to fit geometry-based mesh-quality control into the rules of free energy minimization, providing a cohesive model of flexible membranes. Our model prescribes that pairs of connected vertices are mechanically linked based on a double-barrier (or triple-valley) potential $V_{in}$ as a function of the edge length $l$ between the pairs of vertices (Fig 2B). The dynamics of the vertices can activate two remeshing operations (Fig 2B): i) An increase of the distance between two connected vertices beyond the far barrier leads to splitting of the connection and introduction of a third vertex; and ii) a shortening of the distance below the near barrier leads to removal of the connection and merging of the two vertices into one. Based on the spatially restrictive design of $V_{in}$, iteratively remeshing preserves high mesh quality despite erratic morphodynamics. The method is gradient descent-based and thus exponentially faster than the Metropolis-based searches discussed above. We demonstrate the remeshing performance in several scenarios of strong local changes of cell shapes without compromising the quality of individual triangles (Fig 2C). Overall, we fill a significant gap in the current methodology for modeling free-form membranes by introducing a remeshing that 1) follows the physical rules of free-energy minimization while 2) following geometrical rules that preserve high mesh quality.

In addition, we related the remeshing with another physical process—tension propagation —to study a recent experiment on the mechanical decoupling of membrane tethers [26], *i.e.*

distant but connected tethers losing mechanical signals from each other. This decoupling effect overturns the widely accepted theory of instantaneous tension propagation in real-cell situations. Originally, the decoupling has been interpreted as a result of barriers immobilized by the cytoskeleton that slow the tension transportation. Using the proposed remeshing framework, we have been able to reproduce the decoupled tethers computationally and found that, in addition to the barriers, membrane reservoirs [27] decelerate tension propagation, which causes a further decoupling of the membrane mechanics in distant locations. This example highlights how our proposed remeshing framework empowers simulations that unveil unexpected mechanisms.

The remeshing method is implemented in Matlab. We provide scripts for every application studied as part of the following Results section. The computation of the Helfrich energy is written in C for high speed and integrated into Matlab via the MEX function. The method is supported by Matlab's parallel computing toolbox for large-scale computations.

## Results

We propose a method to realize a flexible meshing of a cell surface in a free energy-dependent model of the membrane. In an initial step, the membrane is meshed with triangles. The sequence of the vertices in each triangle follows the right-hand rule with the normal pointing cell outwards (Fig 2D). The dynamics of the vertices follow the minimization of the total free energy in the membrane system, which is composed of a newly designed internal potential $V_{in}$, the Helfrich energy, and other terms describing the membrane tension, osmotic pressure and external mechanics. If the dynamics cause any barrier crossing in $V_{in}$, the connectivity between the vertices is adjusted by either a splitting- or merging-based remeshing manipulation. Here, we introduce the free energy terms, describe the remeshing, implement the method in five applications of cell morphodynamics and use the method to study the conditions of mechanical decoupling among connected membrane tethers, as experimentally observed in [26].

### Internal potential $V_{in}$

The internal potential is a scalar function of the edge length $l$

$$
\begin{aligned}
V_{in}(l)/V_0 \quad = \quad & \boxed{b\{[1 + e^{-k_{11}(l-r_{11})}] + [1 + e^{-k_{12}(r_{12}-l)}]\}} : \text{Near barrier} \\
+ \quad & \boxed{b\{[1 + e^{-k_{21}(l-r_{21})}] + [1 + e^{-k_{22}(r_{22}-l)}]\}} : \text{Far barrier} \\
+ \quad & \boxed{\tanh[k_w(l - l_{min})] + \tanh[k_w(l - l_{max})]} : 2 \text{ walls} \\
+ \quad & V'.
\end{aligned}
$$

All parameter values and the units used in this paper are indicated in Table 1. The potential depicts two local barriers (exponential terms) and two walls (hyperbolic tangent terms) that regulate the edge length (Fig 2B). The walls forbid the edge length to go outside the bounds $[l_{min}, l_{max}]$. The two barriers further confine the edges within three ranges, which will play a decisive role in the remeshing operations. The term $V'$ is added to offset min $V_{in} = 0$.

In the middle valley between the two barriers (Fig 2B), edges are retained even under moderate perturbation, keeping $l \approx l_0$. Accordingly, the triangles remain nearly equilateral. Strong expansive perturbations can drive the edges into the valley across the far barrier, where $l \approx 2l_0$ is favored. An edge entering this far valley will split into two $\sim l_0$-long edges to restore the geometry. Strong compressive perturbations can drive the edges into the near valley, where $l \approx 0$ is favored. An edge entering this valley will merge into a vertex to restore the geometry. The formulated potential thus imposes geometrical constraints to the edges to neutralize the

**Table 1. Parameter values.**

| Scale | Value | Note |
|---|---|---|
| Energy : $\bar{E}$ | $10 k_B T$ | $T = 300K$ |
| Length : $\bar{l}$ | $1000nm$ | |
| Time : $\bar{s}$ | $1 second$ | |

| Parameter | Value | Reference |
|---|---|---|
| $\kappa$, bending modulus | $\bar{E}$ | same order [35, 36] |
| $C_0$, spontaneous curvature | $0$ | This paper |
| $\mu$, vertex mobility | $100\bar{l}^2\bar{E}^{-1}\bar{s}^{-1}$ | This paper |
| $l_0$, default edge length | $\bar{l}$ | This paper |
| $V_0$, normalization factor of $V_{in}$ | $0.2\bar{E}$ | This paper |
| $b$, barrier height of $V_{in}$ | $0.5$ | This paper |
| $r_{11}$, near slope location of near barrier | $0.05\bar{l}$ | This paper |
| $r_{12}$, far slope location of near barrier | $0.75\bar{l}$ | This paper |
| $r_{21}$, near location of far barrier | $1.25\bar{l}$ | This paper |
| $r_{22}$, far location of far barrier | $1.75\bar{l}$ | This paper |
| $k_{11}$, steepness of near slope of near barrier | $50/\bar{l}$ | This paper |
| $k_{12}$, steepness of far slope of near barrier | $25/\bar{l}$ | This paper |
| $k_{21}$, steepness of near slope of far barrier | $25/\bar{l}$ | This paper |
| $k_{22}$, steepness of far slope of far barrier | $50/\bar{l}$ | This paper |
| $k_w$, steepness of wall | $100/\bar{l}$ | This paper |
| $l_{\min}$, location of near wall | $-0.05\bar{l}$ | This paper |
| $l_{\max}$, location of near wall | $2.2\bar{l}$ | This paper |
| $dl$ | $0.0001\bar{l}$ | This paper |
| $l_{--}$, compressive BCE finish | $0.1\bar{l}$ | This paper |
| $l_-$, compressive BCE trigger | $0.7\bar{l}$ | This paper |
| $l_+$, extensive BCE trigger | $1.3\bar{l}$ | This paper |
| $l_{++}$, extensive BCE finish | $1.7\bar{l}$ | This paper |
| $k_d$, diffusion coefficient | $0.025 - 0.25\bar{s}^{-1}$ | This paper |
| $k_v$, osmotic pressure coefficient | $4\bar{E}\bar{l}^{-3}$ | This paper |
| $k_s$, global tension coefficient | $8\bar{E}\bar{l}^{-2}$ | This paper |
| $k'_s$, local tension coefficient | $12\bar{E}\bar{l}^{-3}$ | This paper |

deteriorations caused by external perturbations. The $V_{in}$-determined dynamics and remeshing can also be interpreted as the spatial rearrangement of lipids responding to the perturbations.

Based on the formulation of $V_{in}$ the free energy term resulting from the internal potential at any vertex $i$ is set to

$$E_{in}(i) = \sum_{j \in nb(i)} \frac{1}{2} V_{in}\left(|\vec{r}_i - \vec{r}_j|\right) \tag{1}$$

where each $V_{in}$ is evenly shared between neighboring vertices; and $V_{in}$ values based on the distances between the position of vertex $i$ and the positions of its 1-ring neighbors (Fig 2E) are summed. Accordingly, the resulting internal force is written as

$$\vec{f}_{in}(i) = -\vec{\nabla}_i E_{in} = -\left[\frac{\partial E_{in}(i)}{\partial r_\alpha(i)} + \sum_{j \in nb(i)} \frac{\partial E_{in}(j)}{\partial r_\alpha(i)}\right]\hat{\alpha}, \tag{2}$$

where Einsteins summation rule is applied to the three Cartesian components indicated by $\alpha$. Of note, $E_{in}(i)$ is governed by the position of vertex $i$ relative to its neighboring vertices $nb(i)$. Shifts in the position of vertex $i$ lead to changes in the local $E_{in}$ of vertex $i$ and all neighboring vertices $nb(i)$. Therefore, there are two partial-derivative terms in $\vec{f}_{in}$, addressing how spatially varying $i$ and $nb(i)$ will change $E_{in}$ and the internal force.

## Other free energy terms

First, the Helfrich energy $H$ determines the membrane resistance to bending. For a closed membrane surface, $H$ depends on the integrated difference between the local mean curvature $C$ and the intrinsic mean curvature of the membrane $C_0$ [28], known as the spontaneous curvature model

$$H = \frac{\kappa}{2} \int_S dS (2C - C_0)^2,$$
(3)

where $S$ denotes a closed-2D surface embedded in 3D space, and $\kappa$ is the bending modulus of the membrane. According to the Gauss-Bonnet theorem [29], the Gaussian curvature term in the bending energy is a constant for closed membranes with $C_0 = 0$, and thus can be ignored for all the applications in our work. The continuous Helfrich free energy is replaced by its discretized version

$$\mathbf{H} = \sum_{i=1}^{N_v} \mathbf{H}(i) = \kappa/2 \sum_{i=1}^{N_v} A_v(i) [2|C(i)|]^2,$$
(4)

where $N_v$ is the total number of vertices; and at the $i$th vertex, the local Helfrich energy $\mathbf{H}(i)$ is governed by the absolute value of the discrete mean curvature and the Voronoi area $A_v(i)$ (Fig 2E). On a mesh surface, the absolute value of the mean curvature can be approximated by $|C(i)| = 1/2|\vec{K}(i)|$. The curvature operator $\vec{K}(i)$ is defined by the cotangent weight method [30]

$$\vec{K}(i) = \frac{1}{2A_v(i)} \sum_{j \in nb(i)} \left( \cot \theta_{ij} + \cot \varphi_{ij} \right) [\vec{r}(i) - \vec{r}(j)].$$
(5)

This method guarantees complete coverage of the meshed surface and is therefore ideal for computing the Helfrich free energy. Given the discrete form $H(i)$ the bending force at any vertex $i$ is determined by

$$\vec{f}_b(i) = -\vec{\nabla}_i H = - \left[ \frac{\partial H(i)}{\partial r_\alpha(i)} + \sum_{j \in nb(i)} \frac{\partial H(j)}{\partial r_\alpha(i)} \right] \hat{\alpha},$$
(6)

applying Einsteins summation rule. As an alternative to this proposed expression of the bending force, the bending resistance could also be derived from the area-difference elasticity (ADE) model [31, 32]. However, due to its overlap with the Helfrich model on reflecting bending, we ignored the ADE model to avoid repetitive computations.

Second, following [33], two additional terms,

$$E_v = k_v \frac{(V - V_0)^2}{V_0}$$
(7)

and

$$E_s = k_s \frac{(S - S_0)^2}{S_0} \tag{8}$$

are implemented to penalize the total volume $V$ and surface area $S$ for diverting from the targeted values $V_0$ and $S_0$ respectively. We applied the method in [34] to compute the three-dimensional volume enclosed by the membrane. $k_v$ and $k_s$ determine the strength of such penalties. The two terms can be interpreted as the influence of osmotic pressure and global membrane tension. The total free energy of the membrane $E$ is thus the sum of $E_{in}$, $\mathbf{H}$, $E_v$ and $E_s$.

## Equations of motion

The forces driving morphology change in any vertex $i$ is determined by the gradient of the free energy terms with respect to the position $\vec{r}(i)$ of the vertex:

$$\vec{f}_{tot}(i) = \vec{f}_{in}(i) + \vec{f}_b(i) + \vec{\xi}(i) - \frac{\partial(E_v + E_s)}{\partial\vec{r}(i)} \tag{9}$$

The potential $V_{in}$ is spatially complex and the corresponding force $\vec{f}_{in}$ is updated adaptively. Whenever the edges reach the potential's complex regions the time step $\Delta t_\delta^{in}$ is recalculated at every simulation step $\delta$. The forces derived from $E_v$ and $E_s$ are averaged over the 1-ring neighbors at each vertex for numerical stability. See Methods and S1 Fig for details.

Based on these considerations, we obtain the equation of motion (EOM) for any vertex $i$ in the membrane as

$$\vec{r}(i, t_\delta) = \vec{r}(i, t_{\delta-1}) + \mu\vec{f}_{tot}(i, t_{\delta-1})\Delta t_\delta^{in}, \tag{10}$$

where $\mu$ is the mobility of the membrane in the Voronoi area around the vertex (Fig 2E).

## Remeshing: Splitting and merging

Given the EOM of the vertices, it is necessary to also dynamically remesh the membrane representation to prevent degeneration of the mesh geometry. Per simulation step, one edge in the mesh may cross a barrier in the potential $V_{in}$, referred to as a barrier crossing event. Such an event can expand an edge by a factor of 2, requiring a split, or compress an edge by a factor of 4, requiring a merge of the two connected vertices (Fig 2B). After a splitting or merging event, all edges are restored approximately to the target length $l_0$. In addition, the connectivities near the added or discarded edges are updated to maintain the triangular mesh topology; a local relaxation is applied to the added or discarded edges to maintain global stability. See the detailed steps of splitting and merging in Methods and S2 Fig. Iteratively applying the remeshing manipulations resolves all barrier crossing events and thus preserves high mesh quality.

## Spontaneous morphodynamics

We applied the EOM/remeshing to simulate the morphodynamics of a biconcave red blood cell (RBC), see Fig 3A. Starting from a sphere, $V_0$ was set to 60% of the volume of the initial sphere, and $S_0$ was set to the surface area of the initial sphere. This constraint imposed a force field that let the morphology spontaneously converge towards a biconcave equilibrium shape (S1 Video). Locally, high mesh quality was preserved despite the global change of the morphology. Obtaining this well-known morphology validates the new method. See [33] for a comparison, where the remeshing is based on geometrical rules.

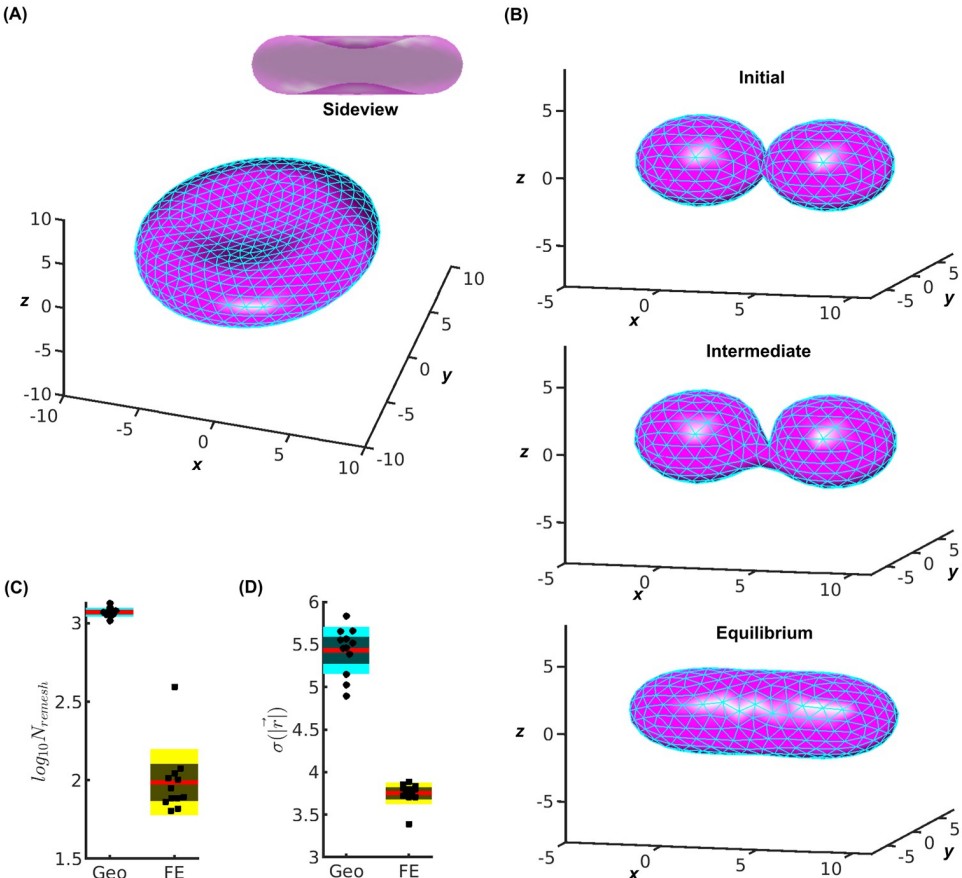

**Fig 3. Morphodynamics of a biconcave red blood cell and a vesicle fusion.** (A) Biconcave morphology at equilibrium ($5 \times 10^4$ iterations). Insert: down-sized side view. The size of the simulated cell is comparable to the typical size of a red blood cell at $6-8\mu m$ in width. See also S1 Video. (B) Vesicle fusion at three stages. $5 \times 10^4$ iterations at equilibirum stage. See also S2 Video. (C) Number of remeshing for geometry- (Geo) and free energy- (FE) based method performing vesicle fusion. (D) Standard deviation of coordinates of remeshing events for Geo- and FE- based method for $2 \times 10^4$ iterations. See also S3 Video.

Next, we applied the EOM/remeshing to simulate the morphodynamics of two fusing vesicles, see Fig 3B. In this example, $V_0$ was equal to the total volume of the two vesicles; and $S_0$ was equal to the total surface area of the two vesicles. Under these assumptions, starting from two vesicles connected with a single triangular mesh, the morphology of the simulated membrane gradually became one elliptical vesicle (S2 Video). High mesh quality was preserved throughout the entire fusion process.

## Advances of free-energy based over geometry-based modeling

We used the simulation of vesicle fusion to compare the performances of the new free energy (FE) and the geometry (Geo) based methods. We chose this example because of its ease of evaluation despite a dramatic morphodynamics. For the Geo-based method, we applied the remeshing purely based on geometry by splitting any edge with $l > l_+$ and merging any edge with $l < l_-$. We broke the relation between remeshing and free energy by setting $V_{in} = 0$. For a fair comparison with the FE-based method, we applied the same local relaxation after each remeshing event in the Geo-based method, which results in equilibration of the edge lengths within the designated range $[l_-, l_+]$.

Although both methods managed to simulate proper fusion (see S3 Video), the Geo-based method required >10 folds more remeshing operations compared to the FE-based method (Fig 3C), although to our surprise, the two methods were similar in computational speed. This was an artifact: Excessive remeshing moves the vertices extensively and equilibrates the membrane fast, which bypasses a part of the free energy-determined Langevin dynamics. However, excessive remeshing operations is detrimental to the accuracy many mesh-derived quantities, including the Helfrich energy. Every remeshing changes the distribution of the 1-ring neighbor number or the valence at certain vertices, which can cause up to 10% difference in Helfrich energy (see areas containing pentagons in Fig 1A).

In addition, Geo-based remeshing happened across the entire membrane whereas FE-based remeshing was concentrated in the connecting area between the two vesicles. Compared to the FE-based method, the standard deviation of the coordinates of the remeshing events $\sigma(|\vec{r}|)$ in the Geo-based method was 2 times larger (Fig 3D). Hence, the FE-based focused the remeshing only to the area of salient morphodynamics where it was necessary. Moreover, the FE-based method resolved mesh irregularity via the spatially restrictive $V_{in}$. In contrast, remeshing events in the Geo-based method, which lacks these constraining mechanisms, occurred in a randomly dispersed fashion.

## Morphodynamics under external forces

Most shape changes are driven by intra/extracellular structures mechanically coupled to the membrane. To reflect how these structures affect the morphodynamics we add an external term to the total free energy. Specifically, we represent these structures by external control points that are connected to select target vertices and their 1-ring neighbors via Hookean springs with a spring constant $k_{ex}$ (Fig 4A). As an approximation of the high rigidity of the mechanical structures relative to the membrane, we fix the position of the control points.

Using the spring forces exerted on the membrane, we mimic several cell morphogenic processes entailing substantial shape changes. Our investigation of the method's performance focused on the relationship between morphological changes and mesh quality.

*Filopodium.* We simulated the formation of this most salient form of cell protrusion, often encountered in neurons and other types of motile cells [37]. To trigger the extension of three narrow tubes from a globular cell, we placed three external control points away from the sphere while maintaining the overall morphology and location with external control points on the sphere (Fig 4B). The equilibrium shape correctly reflected the very local expansion of the membrane without reducing mesh quality throughout the simulation, neither on the tubes or tips nor at the necks (S4 Video).

*Lamellipodium/ruffle.* We simulated the formation of this sheet-like form of cell protrusion, often encountered at the leading edge of polarized, motile cells and around non-polar but stimulated cells [38]. To trigger the extension of a flat membrane-fold from a globular cell, we placed a rim of external control points away from the sphere and further extended these points after half of the simulation while maintaining the overall morphology and location with external control points on the sphere (Fig 4C). The equilibrium shape correctly reflected the expansion of the fold without reducing the mesh quality near the rim or the neck. The method accomplishes this performance by inserting a significant number of vertices on the protruding portion of the membrane, avoiding the build-up of internal tension along the neck (S5 Video).

*Invagination.* We simulated the formation of intracellular invaginations of the cell membrane, often encountered at the surface of cells during entry processes such as endocytosis and macropinocytosis [39]. To trigger the retraction of a membrane dome from a globular cell, we placed a hemisphere of external points inside the cell volume while maintaining the overall

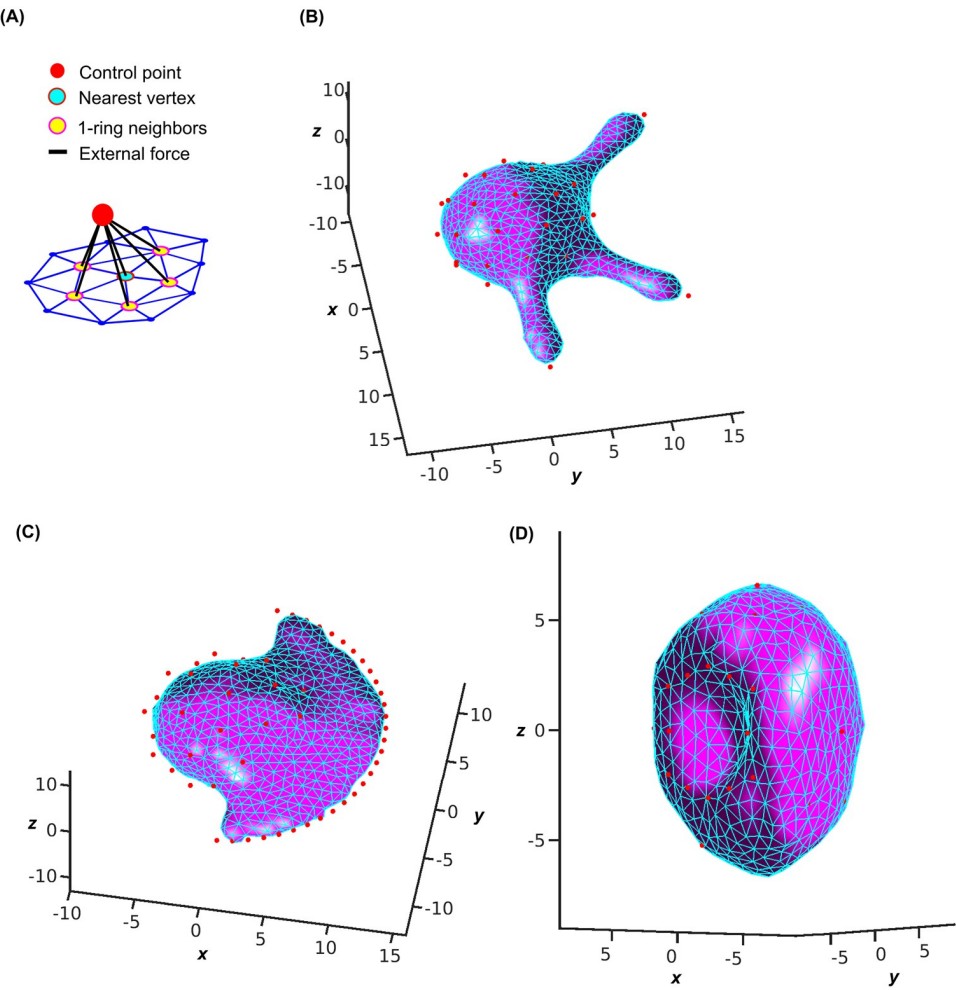

**Fig 4. Applications of the proposed method to generate high-quality meshes of salient geometries under external forces.** (A) Definition of external control points attracting the nearest mesh vertex $i$ and $i$'s 1-ring neighbors $nb(i)$. (B) Formation of a filopodium by placement of a control point at the tip and control points holding the remainder of the globular cell in place. (C) Formation of a lamellipodium by placement of control points in a circular rim and control points holding the remainder of the globular cell in place. (D) Formation of an invagination by placement of a control point 1/3 into the globular cell and control points holding the remainder of the cell in place. Matching videos are provided in S4–S6 Videos ($2 \times 10^4$ iterations).

morphology and location with external points on the spherical surface (Fig 4D). The equilibrium shape correctly reflected the inward curving of the membrane without reducing mesh quality throughout the simulation, neither on the concave dome nor along the convex surrounding area (S6 Video).

## Mechanical decoupling via deceleration of tension propagation

Membrane tension is generally assumed to equilibrate across the entire cell almost instantaneously via diffusion of lipids [40]. However, a recent experiment [26] shows that this assumption is only valid for naive membranes, such as those in artificial giant unilamellar vesicles (GUV) [41] or blebby regions of HeLa cells [26], but not for complex membranes near the cytoskeleton. In Shi *et al.*s experiment, the authors examined the mechanical coupling of two membrane tethers. When attached to a blebby membrane region of a cell, pulling one tether

by an optical tweezer increases tension locally to narrow the tether. The increased tension readily propagates to the other tether, which narrows under the load. Conversely, when attached to a membrane region in contact with the cytoskeleton, pulling one tether only narrows the same tether but not the other. The increased tension is contained to the same tether, which does not narrow the other tether. Compared to the naive membrane, complex membranes are usually penetrated by immobile barriers of cytoskeleton-associated proteins. The authors made the argument that it is the reduction in lipid diffusion due to the barriers, which delays the tension propagation from one tether to the other.

We applied our new method to study this phenomenon quantitatively. The original modeling and interpretation in [26] omitted the spatial geometry of the membrane tethers and made the assumption of extremely slow diffusion of lipids to be able to explain the decoupling. We implemented the geometry and mechanics of the tethers, and successfully reproduced the mechanically coupled tethers of the naive membrane and the mechanically decoupled tethers of the complex membrane containing a diffusion barrier (Fig 5). Additionally, we considered the buffering effect of membrane reservoirs on tension propagation [27, 42], which is not discussed in [26].

We first associated the remeshing with a normalized lipid density $\rho$ to introduce tension propagation. Initially, we set $\rho = 1$ at every vertex. After a splitting event, two vertices become three and their densities are redistributed equally as $\rho = 2/3$ (Fig 5A). Similarly, after a merging event, two vertices become one with $\rho = 2$. We introduced a diffusion-driven vertex-to-vertex exchange of lipids (Fig 5A) by iteratively running the discrete master equation

$$\Delta\rho(i) = \frac{k_d}{v(j)}\rho[j \in nb(i)] - k_d\rho(i) \tag{11}$$

that describes the lipid exchange between a given vertex $i$ and its 1-ring neighbors (Einsteins rule on $j$ implied). For all of our applications, we run Eq (11) for $n_d$ iterations after any remeshing event. The constant $k_d$ defines the rate of the diffusion and $v(j)$ is the number of 1-ring neighbors (or valence) at vertex $j$. Normalization of the positive term in Eq (11) by $v(j)$ equally distributes the outgoing flow from $j$ to all its neighbors, with vertex $i$ receiving only part of the flow. This edge-based diffusion smoothened the abrupt changes to the density caused by the remeshing.

Next, we modified (Eq (8)) controlling global tension to

$$E'_s = k'_s \sum_i \frac{[s(i)/\rho(i) - s_0(i)/\rho_0(i)]^2}{s_0(i)/\rho_0(i)}, \tag{12}$$

where $\rho_0 = 1$ is the initial normalized lipid density at every vertex. Eq (12) describes the summation of locally heterogeneous tensions based on $\rho$, assuming that a higher or lower $\rho$ yields a higher tension after remeshing. As $\rho$ homogenizes under the influence of lipid diffusion (see Eq (11)) these tension fluctuations across the cell surface vanish. Diffusion barriers and membrane reservoirs delay the equilibration, causing a mechanical decoupling of different membrane sites.

Proper practical implementation of this schema required an appropriate compromise between $k_d$ and $n_d$: With large $k_d$ values the density $\rho$ in Eq (11) would change too much causing instability in the morpodynamics associated with uncontrolled fluctuations in the lipid density. Too small $k_d$, on the other hand, would prevent sufficient relaxation of the step changes in lipid density introduced by the remeshing. This could be remedied by increasing $n_d$, however at a high computational cost. We thus chose $n_d = 50$ as an empirically best compromise between computing speed and small enough $k_d$ to guarantee smooth density

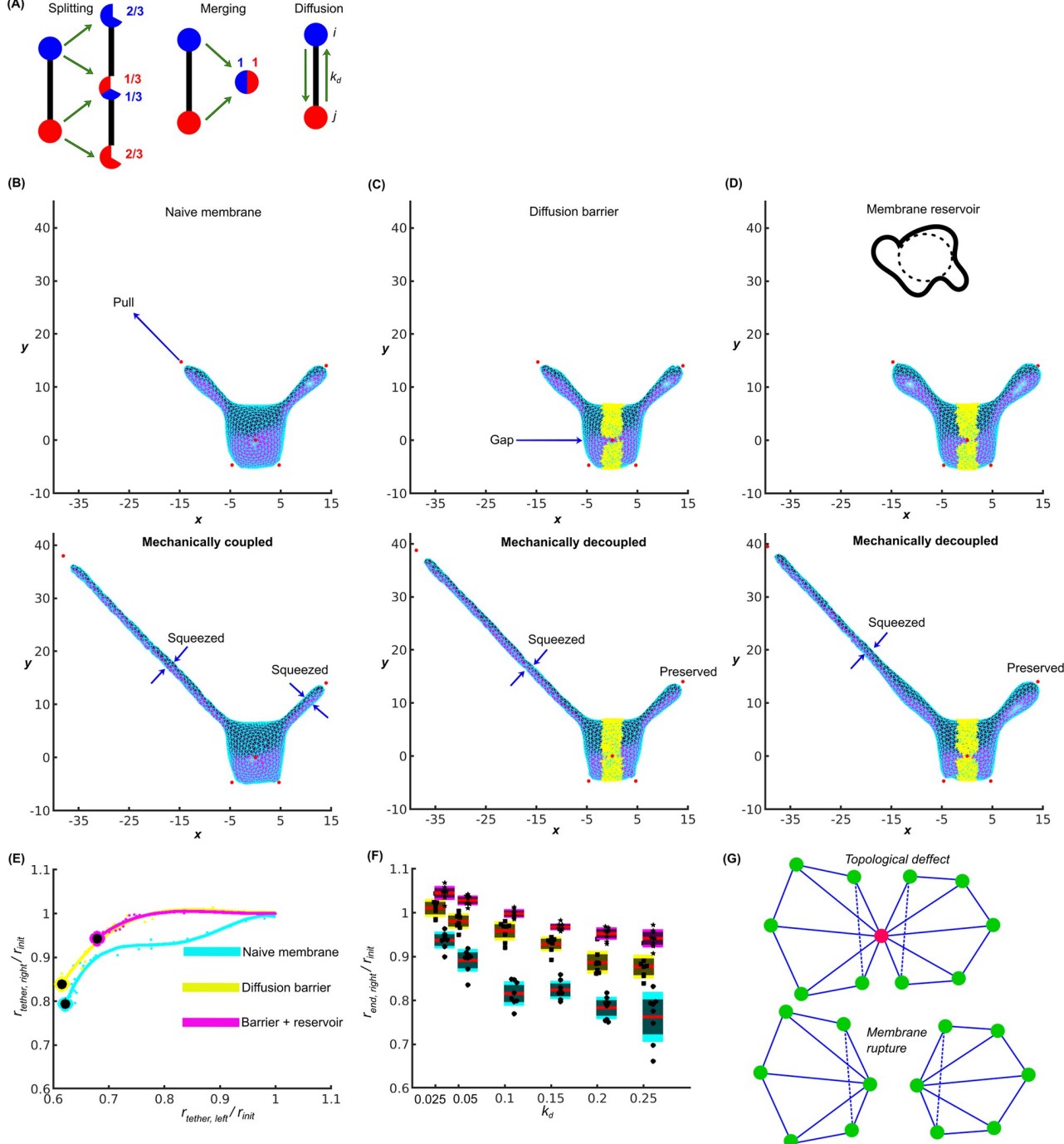

**Fig 5. Applications of the proposed method to study mechanical coupling/decoupling of membrane tethers.** (A) Association of splitting and merging to redistributing lipids, and lipid diffusion between neighbors. (B) Initiation and coupling of two tethers under naive membrane condition. (C) Initiation and coupling of two tethers with diffusion barrier. (D) Initiation and coupling of two tethers with diffusion barrier and membrane reservoir. (E) Fourth-order polynomial fitting of the tether pulling under the three membrane conditions. Radius is normalized to the value of the initial morphology. Circles: last values of radius of left and right tethers obtained from the fitted curves. (F) Scan results of last value of the right tether under six different $k_d$ for the three membrane conditions. (G) Potential topological defect and possible solution.

redistributions after a remeshing step. By stopping the density redistribution after $n_d$ iterations the lipid density stays out of equilibrium, which per Eq (12) drives the experimentally observed mechanical decoupling.

We applied the extended framework to simulate the morphodynamics of tether pulling (Fig 5). To initialize the simulation, we pulled two tethers from a spherical membrane using two external control points. Four additional control points were placed to hold the membrane in place. After this initialization, the membrane reached the equilibrium (Fig 5B). Then, the tether on the left ($-x$ domain) was pulled further away. Our simulations correctly predicted that the additional pull narrows the tether. Moreover, the simulations predicted that the tether on the right ($+x$ domain) was also squeezed to a narrower radius. Thus, the two tethers are mechanically coupled (Fig 5B). Our simulations predicted that slower lipid diffusion in between the two tethers would result in a mechanical decoupling. Diffusion reduction was achieved by introducing a barrier highlighted in yellow in Fig 5C. The edges in the barrier forbade diffusion. To avoid completely stopping diffusion, lipid transport was permitted across 6% of the edges in the barrier. Under these conditions pulling the left tether had no significant effect on the diameter of the right tether(Fig 5C). In a third condition, we introduced excess membrane by setting $s_0$ 33% higher in Eq (12). Our simulations predicted that the right tether would remain mechanically decoupled from the left (Fig 5D), because the excess membrane allowed replenishment of lipids near the pulled left tether. Hence, it was not unnecessary to squeeze the distant right tether.

Finally, we computed simulations under all three membrane conditions for six values of $k_d$, and quantitatively measured the changes to the tethers' radius. For each $k_d$, we repeated the simulations eight times for statistical reliability. We measured the radius of both tethers during the pulling and fit the results with a fourth-order polynomial. The fitted radius of the right tether after the pulling (circles in Fig 5E) indicate the level of mechanical coupling between the two tethers: the lower radius, the stronger is the coupling. Our simulations predicted that for lower $k_d$ values the tethers were decoupled regardless of the membrane condition. For intermediate $k_d$ values, the tethers were coupled in the naive condition and decoupled with the diffusion barrier. The decoupling effect was even stronger in presence of excess membrane. For larger $k_d$ values, the tethers were coupled with or without the barrier. However, with the membrane reservoir, the decoupling effect was maintained. We concluded that combining the diffusion barrier with the membrane reservoir decreases the mechanical coupling. Thus, our simulations provide two complementary mechanisms to the assumption of extremely low lipid diffusion rates [26] to explain the experimentally observed decoupling of membrane tethers.

## Discussion

We present an approach for remeshing membrane representations under the rules of free energy minimization. We introduce a new potential that controls the quality of meshes via splitting and merging in the context of the membrane potential. Of particular importance, this approach supports accurate computing of the bending force, which highly relies on the mesh quality (Fig 1). The approach is also computationally efficient and suitable for many problems related to cell membranes.

### Parameter interpretation

The parameters in the internal potential $V_{in}$ are responsible for the geometrical control over the mesh quality. For this purpose, we determine the baseline values of these parameters such that the two walls of $V_{in}$ remain steep. In addition, we determine $V_0$ to set the height of the barriers properly. For $V_0$s smaller than the baseline value, or lower barriers, the edges are more

**Table 2. Computational cost.**

| Application | Run time | Iteration |
|---|---|---|
| red-blood cell | $\sim 5.8 \times 10^3 s$ | $5 \times 10^4$ |
| vesicle fusion | $\sim 3.7 \times 10^3 s$ | $5 \times 10^4$ |
| filopodia | $\sim 5.5 \times 10^3 s$ | $2 \times 10^4$ |
| lamellipodia | $\sim 6.4 \times 10^3 s$ | $2 \times 10^4$ |
| invagination | $\sim 4.9 \times 10^3 s$ | $2 \times 10^4$ |
| tether pulling | $\sim 20-30$ hours | – |

likely to experience the barrier-crossing followed by splitting and merging. Too frequent remeshing could lead to a rapid increase and decrease of the valence numbers, causing many unsplittable and unmergeable edges in large-force regions. As a result, these abnormal edges halt further remeshing and reduce the flexibility of the membrane. For $V_0$s larger than the baseline value, or higher barriers, the edges are less likely to experience the barrier-crossing. This could lead to highly inert mesh topologies under the baseline external force characterized by $k_{ex}$ and the bending force characterized by $\kappa$. Increasing the external force and decreasing the bending force may remedy stagnation of the membrane under large $V_0$s. We provide a section in the Matlab script for users to adjust the parameters.

## Runtime and memory requirements

- *Run time:* The run time is typically one-two hours, see Table 2, with 12 repeats per run. This level of efficiency is achieved because the remeshing operations follow a gradient-descent minimization of the free energy without additional computation step. In contrast, Metropolis-based methods rely on a large number of random remeshing trials, most of which are eventually discarded [9]. However, because of the much simpler operations associated with each trial the net computation times of our free energy-based and geometry-based methods are comparable. Rather than computational efficiency, the key advance of the proposed method over geometry-based methods is the significantly increased numerical stability, which relates to the application of much fewer but essential remeshing operations. To improve the efficiency for computing larger membranes, adaptive meshes [21] could be applied to areas of different levels of morphological variance, *i.e.* flat areas are represented by large meshes and rugged areas by small meshes. Also, using the proposed remeshing for training, a neuronal network operator can be developed to provide a computationally faster adjustment of the vertex positions and connections to replace the actual remeshing [43].

- *Memory:* The memory cost of our model is mainly on Matlab's infrastructure ($\sim 1.8GB$) and parallel workers ($\sim 660MB$ per worker). Increasing the vertex number, *e.g.* from 162 to 10242 raise the memory cost by only $\sim 10MB$. Additional memory is spent on constructing a Matlab object that hosts the membrane-related variables and functions. Using the Matlab object modularizes the membrane model to enhance the reusability of the model by facilitating the interactions between the model and other components.

## Model limitations

Our method is limited by spatial scales. Obtaining molecular details of the interaction between membrane components and the external environment requires finer treatments such as molecular dynamics or coarse-grain models. A recent publication [44] discusses a framework

to convert meshed membranes to the coarse-grain structures of the basic components of the membrane. When finer details are needed this method could be applied to adaptively convert the mesh results into membrane representations with discrete elements.

Discretizing the Gaussian curvature based on triangular meshes would be numerically unstable, at vertices with either irregular valences [45] or irregular edge lengths [46]. Therefore, we exclude the Gaussian curvature and avoid these numerical instabilities. This choice limits the application of our method to homogeneous membranes with closed topologies. On such membranes, the integration of the Gaussian curvature is a constant, according to the Gauss-Bonnet theorem [29], and thus the bending force unaffected. In an application such as the tether pulling, the mesh may encounter topological defects when the mechanical perturbation on the membrane is overwhelmingly strong, or the morphology becomes too extreme. For example, a tether too thin could remain connected not via triangular elements but a single vertex, see Fig 5G. Although not treated in this work, such a defect could be modeled as membrane rupture and restored after having two separate closed membranes. In addition, because of our neglect of the Gaussian curvature, anisotropic bending that affects the tubulation of the membrane [47–51] cannot be addressed with the current method.

## Extension applications

Although the current model is restricted to spatially homogenous membrane properties, heterogenities in spontaneous curvature and elasticity can still be simulated by adding particle-, mesh- or other coarse-grained-based components [52]. These components can change the morphodynamics through interaction terms related to the membrane in the free energy. These terms could be additive and have no conflict with the original energy terms in the current membrane model. In other words, a heterogenous membrane can be decomposed into a homogenous membrane, treated by the current model, and the other mechanical components, treated by newly designed interactions with the current model. For example, clathrin coats [13], cytoskeletons and proteins with BAR-domain [53] that bend the membrane can be simulated in this manner.

Similarly, we assume that the membrane is externally sculpted by a set of constant point forces. However, it is straightforward to add dynamics to either the external control point distribution location or to the force magnitude an external control point exerts in order to simulate mechanical processes outside or inside the cell and even to capture feedback between membrane shape and external forces. For example, cortical actin network growth or the dynamics of actin filaments in filipodia and lamellipodia are in direct feedback interaction with the morphodynamics of the membrane. Force feedbacks also arise from curvature sensitive signals [54]. The proposed algorithm can capture such relations by spatiotemporal adjustment of external force magnitudes. This will permit integration of the proposed membrane model with additional biochemical and mechanical systems. This possibility highlights the advantage of a meshing framework that couples a physical model to optimized representation of the membrane geometry.

The current single-layer membrane description can readily be replaced by a two-layer description that captures more realistically the double-leaflet structure of membranes. To accomplish this, we propose to extend the current free energy description with an additional term defining the interaction between the two layers. All the other terms in the current model remain unchanged. The two layers will be independently remeshed. The new interaction term will be permissive for lateral sheer motions but penalize orthogonal motions to reflect the fluid-like and hydrophobic features of the membrane. Asymmetric properties could be assigned to each of the two layers [55].

## Methods

We introduce an adaptive Langevin equation (LE) to implement the dynamics of the membrane subject to a remeshing, in which both the mechanically driven motion and the geometric regulation of mesh quality are considered.

### Conventional Langevin equation

The dynamics of the meshed membrane can be described by an over-damped LE that defines the motion of any vertex $i$ by the mechanical coupling to its 1-ring neighbors (Fig 2E), accounting for the bending resistance and attraction from external points, as well as for stochastic forces

$$\frac{d\vec{r}(i)}{dt} = \mu\vec{f}(i) + \sqrt{2\mu k_B T}\vec{R} \tag{13}$$

Here, $\vec{f}$ and $\vec{R}$ denote deterministic and stochastic forces, respectively, and $k_B$ is the Boltzmann constant and $T$ the temperature. The distribution of the random force follows Gaussian white noise. Using natural unit and component expression (using $\alpha$ as Cartesian coordinates, see Table 1), integration of the LE over a finite time step $\Delta t$ yields

$$r_\alpha(i, t + \Delta t) = r_\alpha(i, t) + \mu f_\alpha(i, t)\Delta t + \varepsilon_\alpha(i, t)\sqrt{2\mu k_B T\Delta t} \tag{14}$$

where $\varepsilon_\alpha(i, t)$ is a normal distributed random number with $N(0, 1)$, and $\mu$ is the mobility. The integration of the deterministic force is approximated by multiplication of the time step and the force calculated at the beginning of the time step. However, this simple form of the LE is unsuitable for simulating the membrane dynamics. Unless $\Delta t$ is extremely small, replacing the time integral by a finite step is inaccurate for regions with salient variation in the triple-valley potential (Fig 2B). Thus, we propose an alternative LE, which is based on adaptive time steps to better capture the effect of variation on membrane dynamics.

To be formatically consistent with the deterministic forces, we introduce the time-averaged stochastic force

$$\xi_\alpha(i, t) = \frac{\varepsilon_\alpha(i, t)\sqrt{2\mu k_B T\Delta t}}{\mu\Delta t}, \tag{15}$$

so that the discrete Langevin equation (Eq (14)) is rewritten as

$$r_\alpha(i, t + \Delta t) = r_\alpha(i, t) + \mu f_\alpha(i, t)\Delta t + \mu\xi_\alpha(i, t)\Delta t \tag{16}$$

### Adaptive dynamics

To implement a dynamics that is adaptive to the complexity of the potential, we finely segment regions where $V_{in}$ is complex and coarsen regular regions. Then, we reversely calculate the time steps according to these inhomogeneous segments to refine the simulation. The principle is that every edge, regardless of its location and the complexity of the associated potential, extends or shrinks no more than one segment during each time step. Thus, the edges scan through all the segments one by one without skipping. Because the complex regions have more segments than the regular regions, the dynamics in the complex regions are treated more delicately.

We first evenly sample the potential values separated by a small distance $dl$ in the edge length-$l$ space (S1 Fig),

$$V_{in}(\mathbf{l}) \text{ for } \mathbf{l} := 0, dl, ..., mdl, ..., Mdl, \tag{17}$$

The total number of $dl$ bins is given by $M = l_{\max}dl - 1$. Note that by removing one $dl$, we avoid the singularity in $V_{in}$ at $l = l_{\max}$.

Next, we divide the $l$ space into segments according to $l$. Each segment consists of one or multiple $dl$ in length

$$\mathbf{s} := \mathbf{s}_1 \cup \mathbf{s}_2 \cup, ..., \mathbf{s}_\omega \cup, ..., \cup \mathbf{s}_\Omega, \tag{18}$$

where $\Omega$ is the total number of segments. A given segment $\mathbf{s}_\omega$ reads

$$\mathbf{s}_\omega := (n_{\omega-1}dl, n_\omega dl], \tag{19}$$

where the coefficient $n_\omega$, defines the inclusive upper-bound of the segment, and $n_{\omega-1}$ defines the exclusive lower-bound of the segment (S1 Fig).

The sequential coefficients $n_0 < n_1 < \cdots < n_\omega < ...n_\Omega$ make $l$ space inhomogeneously segmented. The complex regions possess many short segments, and the regular regions possess fewer but longer segments (S1 Fig). To begin the segmentation, we represent the discrete space $l$ with seven critical points (CP), including the two boundaries at $l = 0$ and $l_{\max}-dl$, three local minima (in the three valleys) and two local maxima (on the two barriers) of $V_{in}(l)$. Therefore, there are six initial segments $\mathbf{s} = \mathbf{s}_1 \cup ...\mathbf{s}_6$ separated by the CPs. Then, if any segment $\mathbf{s}_\omega$ in $\mathbf{s}$ violates any of the two conditions,

1. The linear regression fitting of $V_{in}(l)$ within $\mathbf{s}_\omega$ gives $R^2 > 0.5$

2. The standard division of $V_{in}(l)$ within $\mathbf{s}_\omega$ gives $\sigma < 0.001 V_0$

the segment $\mathbf{s}_\omega$ is divided into two segments from its middle. By repeating this operation until all the segments satisfy the two conditions, the final segmentation is obtained. Within every segment, the sampled $V_{in}(l)$ (dots in S1 Fig) is highly linear and uniform as controlled by the two conditions. For example, in S1 Fig, the red segment is long, where $V_{in}(l)$ is highly linear and moderately varying; the black segment is short, where $V_{in}(l)$ is highly nonlinear; and the blue segment is short, where $V_{in}(l)$ is strongly varying.

After the segmentation, we obtain a piece-wise linear substitute for the original potential (S1 Fig):

$$\mathbf{V}_{in}(l) = V_{in}(n_{\omega-1}dl) + \frac{V_{in}(n_\omega dl) - V_{in}(n_{\omega-1}dl)}{(n_\omega - n_{\omega-1})dl}(l - n_{\omega-1}dl), \text{ for } l \in \mathbf{s}_\omega \tag{20}$$

Next, we write the total time of the simulation as consecutive time steps $\delta$,

$$t_\delta = \sum_{\delta'=0}^{\delta} \Delta t_{\delta'}, \text{ for } \delta = 0, 1, ..., N_\delta \tag{21}$$

with $t_0 = 0$ and $\Delta t_0 = 0$. During a given step $\delta$, the time begins at $t_{\delta-1}$ and ends at $t_\delta$. The choice of the time step $\Delta t_\delta$ is based on an adaptive $\Delta t_\delta^{in}$ indicating when to update the internal force $\vec{f}_{in}$.

We obtain the adaptive time step $\Delta t_\delta^{in}$ based on the internal force. Considering an edge $\epsilon$ within the segment $\mathbf{s}_\omega$ at the beginning, i.e. $l_\epsilon(t_{\delta-1}) \in \mathbf{s}_\omega$, $\epsilon$ can be extended or compressed at most to the mid-points of $\mathbf{s}_\omega$'s neighbors $\mathbf{s}_{\omega+1}$ or $\mathbf{s}_{\omega-1}$ at the end of $\delta$. At $t_\delta$ the edge length is

thus either

$$l_\epsilon^+(t_\delta) = \frac{n_{\omega+1}dl + n_\omega dl}{2} \text{ or } l_\epsilon^-(t_\delta) = \frac{n_{\omega-1}dl + n_{\omega-2}dl}{2} \tag{22}$$

and then reversely calculate the two time steps for reaching these lengths.

Considering a given edge $\epsilon$ connecting vertex $i$ and $j$, the deterministic part of its dynamics in the component-wise Langevin equation reads

$$\begin{aligned} r_\alpha(i, t_\delta) &= r_\alpha(i, t_{\delta-1}) + \mu f_\alpha^{tot}(i, t_{\delta-1})\Delta t_\delta^{in} \\ r_\alpha(j, t_\delta) &= r_\alpha(j, t_{\delta-1}) + \mu f_\alpha^{tot}(j, t_{\delta-1})\Delta t_\delta^{in}, \end{aligned} \tag{23}$$

where the total force is the component version of Eq 9. Then, we calculated the difference in squared-length after $\Delta t_\delta^{in}$

$$|l_\epsilon(t_\delta)|^2 - |l_\epsilon(t_{\delta-1})|^2 = \sum_\alpha \left\{ [r_\alpha(j, t_\delta) - r_\alpha(i, t_\delta)]^2 - [r_\alpha(j, t_{\delta-1}) - r_\alpha(i, t_{\delta-1})]^2 \right\}, \tag{24}$$

which is simplified as

$$\begin{aligned} 2\sum_\alpha &[r_\alpha(j, t_{\delta-1}) - r_\alpha(i, t_{\delta-1})][\Delta r_\alpha(j, t_{\delta-1}) - \Delta r_\alpha(i, t_{\delta-1})] \\ &+ \sum_\alpha [\Delta r_\alpha(j, t_{\delta-1}) - \Delta r_\alpha(i, t_{\delta-1})]^2, \end{aligned} \tag{25}$$

where the difference terms read

$$\begin{aligned} \Delta r_\alpha(i, t_{\delta-1}) &= r_\alpha(i, t_\delta) - r_\alpha(i, t_{\delta-1}) \\ \Delta r_\alpha(j, t_{\delta-1}) &= r_\alpha(j, t_\delta) - r_\alpha(j, t_{\delta-1}) \end{aligned} \tag{26}$$

Then, by substituting Eq 26 in Eq 24 and ignoring the second order term we have

$$|l_\epsilon(t_\delta)|^2 - |l_\epsilon(t_{\delta-1})|^2 \approx 2\sum_\alpha [r_\alpha(j, t_{\delta-1}) - r_\alpha(i, t_{\delta-1})][\Delta r_\alpha(j, t_{\delta-1}) - \Delta r_\alpha(i, t_{\delta-1})], \tag{27}$$

Substituting Eq 23, to replace the difference terms in Eq 27

$$|l_\epsilon(t_\delta)|^2 - |l_\epsilon(t_{\delta-1})|^2 \approx 2\mu\Delta t_\delta^{in}\sum_\alpha [r_\alpha(j, t_{\delta-1}) - r_\alpha(i, t_{\delta-1})]\left[f_\alpha^{tot}(j, t_{\delta-1}) - f_\alpha^{tot}(i, t_{\delta-1})\right],$$

Finally, the expression of the adaptive time step for forward (extensive) and backward (compressive) jumping is obtained as

$$\begin{aligned} \Delta t_\delta^{in+}(\epsilon) &\approx \frac{|(n_{\omega+1}dl + n_\omega dl)/2|^2 - |l_\epsilon(t_{\delta-1})|^2}{2\mu\sum_\alpha [r_\alpha(j, t_{\delta-1}) - r_\alpha(i, t_{\delta-1})][f_\alpha^{tot}(j, t_{\delta-1}) - f_\alpha^{tot}(i, t_{\delta-1})]} \\ \Delta t_\delta^{in-}(\epsilon) &\approx \frac{|(n_{\omega-1}dl + n_{\omega-2}dl)/2|^2 - |l_\epsilon(t_{\delta-1})|^2}{2\mu\sum_\alpha [r_\alpha(j, t_{\delta-1}) - r_\alpha(i, t_{\delta-1})][f_\alpha^{tot}(j, t_{\delta-1}) - f_\alpha^{tot}(i, t_{\delta-1})]} \end{aligned} \tag{28}$$

where '+' and '−' indicates extension and compression respectively. Because of the different signs of the numerators but the same denominators, only one deformation gives a positive

time step, which is the physically plausible choice:

$$\Delta t_\delta^{in}(\epsilon) = \begin{cases} \Delta t_\delta^{in+}(\epsilon) & \text{if } \Delta t_\delta^{in+}(\epsilon) > 0 \\ \Delta t_\delta^{in-}(\epsilon) & \text{else.} \end{cases} \quad (29)$$

Each edge triggers a different time step. We choose the shortest step

$$\Delta t_\delta^{in} = \min[\Delta t_\delta^{in}(\epsilon)], \forall \epsilon = 1, ..., \Omega. \quad (30)$$

$\Delta t_\delta^{in}$ ensures that the edge under maximal extension or compression crosses only one segment. Deformations of all edges are limited to one segment or less allowing rapid computation of the updated potential via the piece-wise linear approximation. Therefore, the adaptive LE (Eq 10) based on the adaptive $\Delta t_\delta^{in}$ optimally captures the spatial complexity of the potential $V_{in}$.

## Steps of splitting and merging

Following a barrier crossing event (BCE) in $V_{in}$, we outline the steps S1–S5 of a split, and steps M1–M5 of a merge (S2 Fig), starting from the initial mesh configuration shown in (S0) and (M0) respectively.

- *Splitting:*
  (S1) Given an edge $\epsilon$ connecting vertices $i_1$ and $i_2$ with a length $l_\epsilon > l_+$, the global dynamics of the mesh is paused to allow relocation of only $i_1$, $i_2$ and their 1-ring neighbors (green circles). These vertices respond only to the internal forces and noise. The bending and the external forces are halted to avoid inaccuracy in computing these forces caused by the extended $\epsilon$. Additionally, we include the positions of $\epsilon$'s 2-ring neighbors as a fixed boundary (blue dots). In this way, we can complete the BCE while limiting the effects of changes in local mesh topology to the other vertices, hereon refered as the local relaxation.
  In order to accelerate the local relaxation, we change the Gaussian noise to a biased Levy noise [56] for efficiently extending $\epsilon$ and keeping the other edges near $l_0$. The Levy noise is biased to extend $\epsilon$ and nearby edges that get overly short ($l < l_-$) or overly long ($l > l_+$) are stabilized. Two possible results are expected. First, if $l_\epsilon > l_{++}$ and the other edges satisfy $l_- < l < l_+$, the local relaxation is complete and the BCE is successful. The splitting proceeds to the next step (S2). Second, if $l_\epsilon < l_+$ and the other edges satisfy $l_- < l < l_+$, the local relaxation is also complete but the BCE is unsuccessful. The splitting is canceled and the mesh is restored to the initial configuration (S0). Note that the local relaxation merely provides an intermediate mesh configuration to facilitate the remeshing. The final shape of the membrane is determined by the global dynamics with all the forces restored after finishing the remeshing.
  Before describing the following steps, we clarify the notation: the vertices connecting to both $i_1$ and $i_2$ are numbered as $j_1$ and $j_2$. Vertex $j_1$ fulfills the right-hand rule in triangle $j_1 \rightarrow i_1 \rightarrow i_2$ (Fig 3A), vertex $j_2$ fulfills the right-hand rule in triangle $j_2 \rightarrow i_2 \rightarrow i_1$. Vertex $k_1$ forms a triangle with the edge $i_1 j_1$, and analogously, vertices $k_2$, $k_3$, and $k_4$ form triangles with the edges $i_2 j_1$, $i_2 j_2$ and $i_1 j_2$, respectively.
  (S2) To split $\epsilon$ we insert vertex $i_n$ at the mid-point between $i_1$ and $i_2$, yielding two $\sim l_0$-long edges. Then, we define a new mesh structure around $i_n$ by disconnecting $i_1 i_2$, $i_1 j_1$, $i_2 j_2$, $i_1 j_1$ and $i_2 j_2$ and connecting $i_n$ to its closest neighbors, i.e. $i_n i_1$, $i_n i_2$, $i_n j_1$ and $i_n j_2$. The outcome of this step is four quadrilaterals, which require proper triangulation.
  (S3) Each quadrilateral has two options (magenta dashed lines) for triangulation, yielding $2^4 = 16$ options of triangular meshes. We only keep options that preserve the valence of every vertex between 5 and 8. The valence of a given vertex is equal to the number of its 1-ring neighbors. The choice between 5 and 8 avoids unreasonable geometries. If no option

can satisfy the valence requirement, the edge is considered unsplittable, and the configuration restored to (S0). Additionally, the extended $\epsilon$ is restored to $l_\epsilon < l_+$ via a similar local relaxation as in (S1), except that the Levy noise in this case compresses $\epsilon$. Otherwise, if one or more options can satisfy the valence requirement, the option minimizing the variance of the valence is chosen.

(S4) The new edges (black dashed lines) in (S3) undergo a second local relaxation, similar to (S1). All of the moveable vertices in the new edges experience the internal force and Levy noise. The neighbors at the local border remain fixed.

(S5) The local relaxation is complete when every edge satisfies $l_- < l < l_+$.

- *Merging:*

  (M1) Given an edge $\epsilon$ connecting vertices $i_1$ and $i_2$ and a length of $l_\epsilon$, when $l_\epsilon < l_+$, the global dynamics of the mesh is paused to allow relocation of only $i_1$, $i_2$ and their 1-ring neighbors (green circles). These vertices respond only to the internal forces and noise. The bending and the external forces are halted to avoid inaccuracy in computing these forces caused by the compressed $\epsilon$. Additionally, we include the positions of $\epsilon$'s 2-ring neighbors as a fixed boundary (blue dots).

  In order to accelerate the local relaxation, we change the Gaussian noise to a biased Levy noise [56] for efficiently compressing $\epsilon$ and keeping the other edges near $l_0$. The Levy noise is biased to compress $\epsilon$ and nearby edges that get overly short ($l < l_-$) or overly long ($l > l_+$) are stabilized. Two possible results are expected. First, if $l_\epsilon < l_{--}$ and the other edges satisfy $l_- < l < l_+$, the local relaxation is complete and the BCE is successful; then the merging proceeds to the next step (M2). Second, if $l_\epsilon > l_-$ and the other edges satisfy $l_- < l < l_+$, the local relaxation is also complete but the BCE is unsuccessful; then the merging is canceled and the mesh is restored to the initial configuration (M0).

  Before describing the following steps, we clarify the notation: the vertices connecting to both $i_1$ and $i_2$ are numbered as $j_1$ and $j_2$. Vertex $j_1$ fulfills the right-hand rule in triangle $j_1 \to i_1 \to i_2$ (Fig 3A), vertex $j_2$ fulfills the right-hand rule in triangle $j_2 \to i_2 \to i_1$. Vertex $k_1$ forms a triangle with the edge $i_1 j_1$, and analogously, vertices $k_2$, $k_3$, and $k_4$ form triangles with the edges $i_2 j_1$, $i_2 j_2$ and $i_1 j_2$, respectively.

  (M2) To merge $\epsilon$ we remove $i_2$ and the original edges $i_1 i_2$, $i_1 j_1$, $i_1 j_2$, $i_2 j_1$ and $i_2 j_2$. The outcome of this step is two quadrilaterals, which require proper triangulation.

  (M3) Each quadrilateral has two options (magenta dashed lines) for triangulation, yielding $2^2 = 4$ options of triangular meshes. We only keep options that preserve the valence of every vertex between 5 and 8. If no option can satisfy the valence requirement, the edge is considered unmergable, and the configuration restored to (M0). Additionally, the compressed $\epsilon$ is restored to $l_\epsilon > l_-$ via a similar local relaxation as in (M1), except that the Levy noise in this case extends $\epsilon$. Otherwise, if one or more options can satisfy the valence requirement, the option minimizing the variance of the valence is chosen.

  (M4) The new edges (black dashed lines) in (M3) undergo a second local relaxation, similar to (M1). All of the moveable vertices in the new edges experience the internal force and Levy noise. The neighbors at the local border remain fixed.

  (M5) The local relaxation is complete when every edge satisfies $l_- < l < l_+$.

## Pseudocode

1. Generate an initial mesh.

2. Compute the effective time step $\Delta t_\delta^{in}$ for the current simulation step $\delta$.

3. Advance the dynamics by $\Delta t_\delta^{in}$ following the adaptive Langevin Equation.

4. Pause the dynamics if an edge reaches $l < l_-$ or $l > l_+$, apply splitting and/or merging

5. Resume the global dynamics

6. Return to '2' to repeat 2–5 untill reaching equilibrium.

## Supporting information

**S1 Fig. Linear segmentation of $V_{in}$.** $l$-space discretization and segmentation according to $V_{in}$: examples of complex regions of $V_{in}$ in black (non-linear) and blue (highly varying) segments and regular regions of $V_{in}$ in red segments.
(TIF)

**S2 Fig. Edge splitting and merging steps.** Initial configuration (S0) and steps of splitting (S1-S5); and initial configuration (M0) and steps of merging (M1-M5). See text for detailed procedures.
(TIF)

**S1 Video. Morphodynamics of bi-concave shape formation starting from a sphere.** Video matching Fig 3A.
(GIF)

**S2 Video. Morphodynamics of vesicle fusion.** Video matching Fig 3B.
(GIF)

**S3 Video. Comparison of geometry (Geo)-based and free energy (FE)-based methods.** Green edges indicate edges undergoing remeshing. Video matching Fig 3C and 3D.
(MP4)

**S4 Video. Morphodynamics of invagination.** Video matching Fig 4B.
(GIF)

**S5 Video. Morphodynamics of lamellipodium formation.** Video matching Fig 4C.
(GIF)

**S6 Video. Morphodynamics of invagination.** Video matching Fig 4D.
(GIF)

## Acknowledgments

We thank Sandra Schmid for helpful discussion and comments on our manuscript, and Jenny (Qiongjing) Zou for software management.

## Author Contributions

**Conceptualization:** Xinxin Wang.

**Data curation:** Xinxin Wang.

**Formal analysis:** Xinxin Wang.

**Funding acquisition:** Gaudenz Danuser.

**Investigation:** Xinxin Wang, Gaudenz Danuser.

**Methodology:** Xinxin Wang.

**Project administration:** Gaudenz Danuser.

**Software:** Xinxin Wang.

**Supervision:** Gaudenz Danuser.

**Validation:** Xinxin Wang.

**Visualization:** Xinxin Wang.

**Writing – original draft:** Xinxin Wang, Gaudenz Danuser.

**Writing – review & editing:** Xinxin Wang, Gaudenz Danuser.

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
