## [Decision Letter · Decision Letter 0]

16 Aug 2022

Dear Dr. Danuser,

Thank you very much for submitting your manuscript "Remeshing flexible membranes under the control of free energy" for consideration at PLOS Computational Biology.

As with all papers reviewed by the journal, your manuscript was reviewed by members of the editorial board and by several independent reviewers. In light of the reviews (below this email), we would like to invite the resubmission of a significantly-revised version that takes into account the reviewers' comments.

We cannot make any decision about publication until we have seen the revised manuscript and your response to the reviewers' comments. Your revised manuscript is also likely to be sent to reviewers for further evaluation.

Sincerely,

Nir Gov

Academic Editor

PLOS Computational Biology

Jason Haugh

Section Editor

PLOS Computational Biology

Reviewer's Responses to Questions

**Comments to the Authors:**

Reviewer #1: The following comments are offered to the authors' consideration:

1.page 2, bottom paragraph: The authors wrote that “the computation of the Helfrich free energy is written in C ….”.

Comment: In various Monte-Carlo (MC) and other types of computer programs (including the commercially available computer programs like it is Surface Evolver ) the Gaussian bending term or alternatively the deviatoric term (see for example J. Phys. A: Math. Gen., 38: 8527–8536, 2005) cannot be accurately included in the expression of the membrane free energy since it is numerically difficult to determine the Gaussian curvature and also the curvature deviator. The numerical determination of the local mean curvature C (as you have it in Eq. 3) is much easier. The Gaussian term in Helfrich energy can be omitted for closed membrane shapes by applying the Gauss-Bonnet theorem (usually for spherical topology) and assuming that the membrane is homogeneous (which is in general not true since the membrane constituents are not homogeneously distributed, see for example V. S. Markin: Lateral organization of membranes and cell shapes, Biophys J., 36(1): 1-19, 1981).

The Gaussian bending term was omitted also in the present work, i.e. in the Helfrich bending energy expression the Gaussian term is not taken into account (Eq. 3 in the manuscript) even it is important and cannot be neglected for non-homogeneous membranes, membrane patches, etc.

Further, for the mechanical stability of the highly curved tubular structures (like they are for example the membrane tethers considered in the present manuscript) the anisotropic part of the membrane bending energy (i.e. deviatoric energy term) is important. The anisotropic bending term is the function of the curvature deviator (which can expressed according to Euler formula as a function of the mean curvature and the Gaussian curvature). So also the curvature deviator cannot be taken into account within the method/program presented in this manuscript. See for example Bobrovska et al., PLOS ONE, 8 e73941, 2013 or Kralj-Iglic et al., J. Phys. A: Math. Gen. 35: 1533-1549 , 2002; Fournier, Phys. Rev. Lett. 76, 4436, 1996; Kralj-Iglic et al., Eur. Phys. J. B., 10: 5-8, 1999. Please discuss this issue in the discussion part of the manuscript and also in Introduction.

2.Regarding the relation between the spontaneous curvature model and so-called area-difference elasticity (the authors cited ref. 28 which is not original with respect to these two models), both models are connected, especially in the case of small area difference, see for example Proc. Natl. Acad. Sci. USA, 99: 16766–16769, 2002. So, please discuss also this issue.

3.In analytical approach (calculus of variation) the Eqs. (7) and (8) would be global constraints (for cell volume and cell membrane area) in the functional, while k_v and k_s would be the global Lagrange multipliers whose values (determined in variation of the functional by solving Euler-Lagrange differential equations and separately taking into account also the constraint) would depend on the values of the target volume (V_o) and the target area (S_o). Within this approach k_v and k_s are not just the constants describing the mechanical properties of the membrane and cell (see for example Ou-Yang Zhong-Can et al.: Geometric Methods in the Elastic Theory of Membranes in Liquid Crystal Phases, World Scientific, Singapore). Please discuss in the manuscript also this issue.

4.In the present manuscript, the membrane was sculpted externally by setting points external to the bulk of the membrane in order to mimic the tether pulling that is crucial for the study of tension propagation. Even though this is a somewhat artificial constraint, it reflects the corresponding force-curvature feedback mechanisms mentioned in the text. Protruding forces in many cellular processes are recruited through curved proteins, or complexes containing the curved proteins, for example due to actin polymerization. Here, a reference to relevant work should be added, for example the article published in Soft Matter 15, 5319-5330, 2019 or some other article. How could the authors implement such reciprocal mechanism, where the curvature-sensing membrane inclusions produce a protruding force, in the presented work? Is an addition of membrane inclusions a feasible addition to the model?

5. Could the present model be used to simulate asymmetric membranes in which the mechanics of the outer leaflet differ from the inner one (see Hossein and Deserno, Spontaneous curvature, differential stress, and bending modulus of asymmetric lipid membranes, Biophysical Journal, 118:, 624-642, 2022) and if not, how could this be implemented?

6. The re-meshing mechanics is well described and the three-valley potential is a good idea to assure unbiased meshes. However, the authors could stress more clearly the advantages of their approach to other methods. Is not the re-meshing much more computationally demanding? Furthermore, the authors comment that the method is limited by scaling. How could this be overcome?      

7. In the discussion of limitations of current approaches the authors mentioned several authors, however the work of a few more groups could be mentioned within the text and their work included in the discussions. As for example: Gompper and Kroll, “Triangulated-surface models of fluctuating membranes, in: Statistical Mechanics of Membranes and Surfaces, (pp.359-426, World Scientific (Editors: D. Nelson, T. Piran, S. Weinberg); Kroll and Gompper, “The conformation of fluid membranes - Monte-Carlo simulations,” Science, 255: 968-971, 1992; Penic et al., “Bending elasticity of vesicle membranes studied by Monte Carlo simulations of vesicle thermal shape fluctuations,” Soft Matter, 11: 5004 - 5009, 2015; etc.

8.The authors gave the estimation of the computer time for parameter sweep running on a 12 cores, 256GB RAM computer. Besides wall clock metrics of the complexity of the simulations, it would be beneficial to mention how many time steps there were required to achieve the equilibrium states and the size of the mesh, i.e. number of vertices in the mesh (maybe a final vertex number where the number of vertices vary). With the additional data it will be possible to estimate the computational complexity of a single time step of the simulations.

9. Please explain how much RAM was actually used by simulations? The authors mention only the amount of RAM available on the machine, but not how much RAM was needed by their program. Again, the information about the number of vertices would be also beneficial, since larger vertices means more RAM requirements.

Reviewer #2: The paper by Wang and Danuser presents a new methodology for meshing

triangulated membranes. The new approach combines elements of both

free-energy and geometry-dependent methods for remeshing dynamic

surfaces in computer simulations of processes involving significant

morphological changes. The paper, which is well written, has

introduced me to a set of important technical issues that I was not aware

of in membrane simulations. I believe that it fits the goals of PLOS

Computational Biology, and my overall opinion on the work is

positive.

I would like to ask the authors to clarify the following technical and

conceptual issues:

1. The authors state that their method "preserves mesh quality" on the

one hand (like geometry-based simulations) but do so in a free-energy

regulated manner. This is indeed the impression the one gets from the

nice examples that they provide throughout the paper, but I was

wondering if they can give a specific example where other methods fail

and the newly presented one works well. Can the author specify, for

instance, which of the processes described in the paper will not be

simulated well by other existing methods?

2. Please also discuss the computational toll of the proposed

simulations. In particular, please discuss whether it is comparable to

existing popular schemes?

3. The internal energy in Eq. (1) depends on the the number of edges

of the triangulated surface. Thus, I would expect that it changes when

the surface undergo splitting/merging. It is not clear to me if this

energy change, which seems artificial, has influence on the

dynamics. Please clarify.

4. Related to the above point: In the "three-valley" energy landscape

in fig. 2(B), the minimum of the energy is $V_{in}=3$. Is there a

reason why the minimum in not shifted to zero? Do the absolute values

have importance or only the energy difference between the valleys and

the hills matter?

5. I found the presentation of the results of Figure 5 confusing.

(a) If I understand correctly, the rate of lipid diffusion is

controlled by the constant $k_d$ in Eq. (11). This equation is

"empirically" iterated 50 times after remeshing. Can the authors

rationalize this choice? Does it mean that the local densities do not

continue to change until the next remeshing event?

(b) I did not understand the idea of a "wide gap" in Fig. 5(B). How is

it implemented and how does it hamper diffusion between the two

tethers?

(c) Same for Fig. 5(C). How is this idea implemented? Can the authors

expand on why the excess membrane decouples between the two tethers,

**Have the authors made all data and (if applicable) computational code underlying the findings in their manuscript fully available?**

Reviewer #1: **No: **already explained in the comments to the authors

Reviewer #2: Yes

PLOS authors have the option to publish the peer review history of their article (what does this mean?). If published, this will include your full peer review and any attached files.

Reviewer #1: No

Reviewer #2: No
---

## [Decision Letter · Decision Letter 1]

28 Nov 2022

Dear Dr. Danuser,

We are pleased to inform you that your manuscript 'Remeshing flexible membranes under the control of free energy' has been provisionally accepted for publication in PLOS Computational Biology.

Best regards,

Nir Gov

Academic Editor

PLOS Computational Biology

Jason Haugh

Section Editor

PLOS Computational Biology

Reviewer's Responses to Questions

**Comments to the Authors:**

Reviewer #1: the revised version of the manuscript can be published in its present form

Reviewer #2: I have no further comments.

**Have the authors made all data and (if applicable) computational code underlying the findings in their manuscript fully available?**

Reviewer #1: Yes

Reviewer #2: Yes

PLOS authors have the option to publish the peer review history of their article (what does this mean?). If published, this will include your full peer review and any attached files.

Reviewer #1: No

Reviewer #2: No

---

## [Editor Report · Acceptance letter]

1 Dec 2022

PCOMPBIOL-D-22-01135R1 

Remeshing flexible membranes under the control of free energy

Dear Dr Danuser,

I am pleased to inform you that your manuscript has been formally accepted for publication in PLOS Computational Biology. Your manuscript is now with our production department and you will be notified of the publication date in due course.

With kind regards,

Zsofia Freund
